# A Survey of Full-Cycle Cross-Modal Retrieval: From a Representation Learning Perspective

**Suping Wang [1], Ligu Zhu [1,2,\*], Lei Shi [1,\*], Hao Mo [1] and Songfu Tan [1]**

[1] State Key Laboratory of Media Convergence and Communication, Communication University of China, Beijing 100024, China
[2] Beijing Key Laboratory of Big Data in Security & Protection Industry, Beijing 100024, China
\* Correspondence: zhuligu@cuc.edu.cn (L.Z.); leiky_shi@cuc.edu.cn (L.S.)

**Abstract:** Cross-modal retrieval aims to elucidate information fusion, imitate human learning, and advance the field. Although previous reviews have primarily focused on binary and real-value coding methods, there is a scarcity of techniques grounded in deep representation learning. In this paper, we concentrated on harmonizing cross-modal representation learning and the full-cycle modeling of high-level semantic associations between vision and language, diverging from traditional statistical methods. We systematically categorized and summarized the challenges and open issues in implementing current technologies and investigated the pipeline of cross-modal retrieval, including pre-processing, feature engineering, pre-training tasks, encoding, cross-modal interaction, decoding, model optimization, and a unified architecture. Furthermore, we propose benchmark datasets and evaluation metrics to assist researchers in keeping pace with cross-modal retrieval advancements. By incorporating recent innovative works, we offer a perspective on potential advancements in cross-modal retrieval.

**Keywords:** cross-modal retrieval; representation learning; full-cycle modeling; feature engineering; pre-training tasks

## 1. Introduction

Cross-model retrieval focuses on retrieving information from multiple modalities. In the era of intelligent media, emerging social networking sites are gradually becoming more popular, increasing users' expectations for search results. People's demand for information retrieval is no longer satisfied with a single modality, but they want to obtain data from different modalities. Both domestically and globally, academic scholars are committed to exploring the semantic association between cross-modalities to improve the precision and efficiency of retrievals. Cross-modal retrieval faces severe challenges in modal feature representation, complex semantic processing, the alignment of different modal features, and dataset construction. One of the primary motivations for cross-model retrieval is to bridge the gap between the various modalities and enable effective information retrieval.

Representation learning is the foundation of cross-modal retrieval. It represents and summarizes the complementarity and redundancy of vision and language. Cross-modal representation in our work explores feature learning and cross-modal interactions for information integration. It intends to minimize redundancy across modalities to provide a more effective feature representation. Cross-modal retrieval explores the mechanisms involved in the transformation of knowledge and seeks to maintain semantic consistency across diverse modalities within the context.

Traditional surveys cover various modalities, including text, image, audio, and video modalities. There are also more focused reviews of cross-modal retrieval oriented toward image text. According to the variations in developing inter-modal association methods, traditional studies divided text retrieval approaches into typical association analysis, deep

learning, and deep hashing. They investigated particular remedies to the flaws of cross-modal association methodologies. Kaur et al. [1] compared graphic retrieval surveys and analyzed the practical applications. There is low-level representational heterogeneity and high-level semantic homogeneity between vision and language. Multi-learning systems and explored approaches [2] for fusing representations are developed in deep learning architectures. Cross-modal retrieval attempts to realize information interactions by mining the relationships of multiple modal samples. There are more challenges in effective indexing and retrieval. Feng et al. [3] divided cross-modal modeling strategies into direct and indirect modeling. The former directly measured the correlation between different modal data by establishing a sharing layer. The latter established a semantic correlation between other modals in various scenes by building a common representation space. Wang et al. [4] classified existing cross-modal retrieval methods into real-value and binary representation learning and summarized their respective core ideas. However, massive innovative and high-tech breakthroughs have emerged in recent years. Researchers may abandon past machine learning techniques in the deep learning era. Peng et al. [5] paid attention to cross-media analysis and reasoning, which are not carried out by retrieval. With the advancement of cross-modal retrieval, a comparatively high number of review articles emerged. Cross-modal retrieval [6] introduced public space learning and similarity measurement, and different cross-modal retrieval techniques were summarized. Some researchers divide cross-modal retrieval into subspace-based, deep-learning-based, hash-transform-based, and theme-based methods [7]. The main problem is the lack of research on the association between intra-modal local data and inter-modal semantic structures. Li et al. [8] reviewed cross-modal retrieval models based on representation learning from two dimensions of information extraction. They also provided a summary of feature extraction results. As graph-related knowledge adds to the retrieval process, the cross-modal retrieval method [9] of joint graph regularization was explored. Ayyavaraiah and Venkateswarlu [10] provided a concise overview of the advancements in cross-modal feature retrieval and optimization. Furthermore, they elucidated the prevalent issues and obstacles encountered in data joint analysis.

Compared to preliminary studies, our survey from a representation learning perspective involves full-cycle methodologies that include feature engineering, representation learning, cross-modal interaction, and constructing high-dimensional correlations. The typical survey of cross-modal retrieval is primarily on whether a deep learning model is employed for classification, and then, the retrieval results are analyzed. Different from the traditional works, we focused on the characteristics of deep learning. Taking advantage of pre-training and fine-tuning modes, our survey covers crucial processes of representation, translation, alignment, fusion, co-learning, and cross-modal research.

We provide a taxonomy of issues and challenges on the subject to help readers better understand image text and text image retrieval. The survey thoroughly explains how to overcome recent technological breakthroughs. We examined the problems and challenges that must be solved. Figure 1 contains several issues that require addressing. Detailed introductions are provided in the subsequent sections:

1. Multi-modal data volume: Uni-modal retrieval cannot keep up with the increasing expansion of multi-modal data.
2. Significant differences in the manifestation of heterogeneity: There is an issue with evaluating the content correlation between modal data and computing their similarity.
3. Semantic gaps: The bottom-up semantics in feature analysis between separate modalities are called semantic gaps.
4. Scalability of deep learning models: Adding new datasets, retraining the model, and recalculating all take a long period and need incremental learning.
5. Problem with training datasets: There are missing data, loose data, fewer data labels, and noisy data. Because the volume of particular cross-modal retrieval datasets is now relatively tiny, multi-modal retrieval datasets that are large in scale and a universal representation must be collected.

6.  Granularity of cross-modal correlation: Researchers must connect information at the fine-grained and coarse-grained levels and pay close attention to contextual information.
7.  Text length issue: The textual duration in cross-modal retrieval may be too lengthy or too short to prevent it from expressing complete meaning.
8.  Long tail of vision: Raw data in the vision domain often follow a long-tail distribution, with most samples originating from only a small number of classes.
9.  Intra-modal reasoning problem: The fine-grained information in the modal is semantically dependent, posing an intra-modal reasoning challenge.
10. Inter-modal alignment issue: Fine-grained information alignment and fragment alignment are examples of inter-modal alignment issues.
11. Scarce memory resources: When dealing with vast amounts of data, real-valued representation techniques suffer from expensive computing costs and great space requirements.
12. Large latency and low efficiency in retrieval: Extracting region features or other characteristics might be time-consuming, resulting in delayed retrieval results.

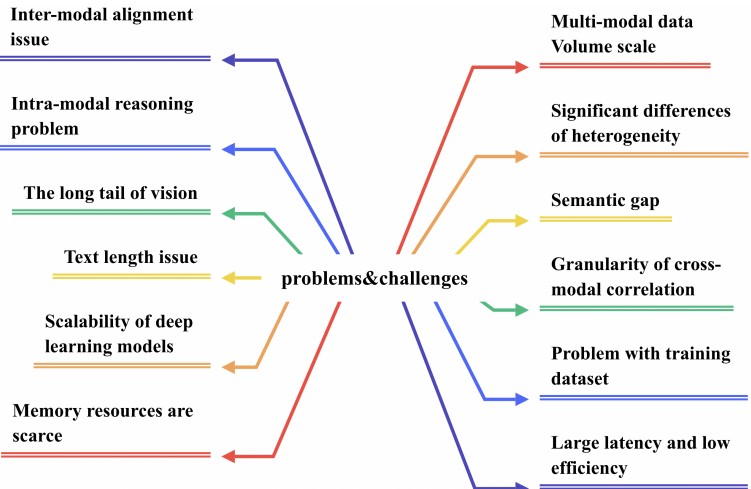

**Figure 1.** The problems and challenges in cross-modal retrieval.

In addition to covering real-value and binary representation methods, our survey further summarizes deep learning methodologies. Compared to previous work, we summarize the overall workflow of cross-modal retrieval, explore the issues and challenges, and analyze the optimization characteristics in each phase. Our contributions are as follows:

- We summarize various open issues and challenges.
- We concentrated on methodologies with a full-cycle deep learning process, which addresses a gap in existing works. Our approach incorporates innovative techniques and references that are absent in prior surveys.
- This paper provides a comprehensive summary and comparative analysis of disparate cross-modal representations at every pre-training stage.
- We present a comprehensive description of benchmark datasets and evaluation metrics that are critical.

The road map is arranged as follows: Section 2 provides preliminary techniques and the retrieval pipeline. Section 3 describes full-cycle methodologies oriented toward fine-grained deep learning, including feature engineering, cross-modal interaction, pre-training tasks, and unified vision language architecture. Feature engineering combines extraction capacity, representation learning, and the high-dimensional correlation of diverse modalities. We analyzed aligning, reconstructing, and embedding the distinct modal information elements. Besides, the survey presents the glossary of pre-training tasks for cross-modal information completion and matching. Section 4 compares optimal studies about loss

functions, avoiding the problems of gradient exploding or vanishing in detail. Additional performance metrics with various evaluation methods are demonstrated in Section 5. Section 6 illustrates different benchmark datasets in cross-modal retrieval. Comparison results of cross-modal representation on the Flickr 30k and MS-COCO datasets are also provided. Finally, Section 7 concludes the survey.

## 2. Overview of Cross-Modal Retrieval

Deep learning models have advanced the domain of cross-modal retrieval by addressing the heterogeneity challenge across diverse modalities. These approaches predominantly emphasize techniques including joint subspace learning, feature extraction, interaction, alignment, matching, and fusion. A considerable number of researchers establish fine-grained interactions and multi-level or multi-stage semantic alignments to mitigate the disparities between visual and linguistic modalities. Furthermore, numerous image text pair corpora can be fine-tuned to perform retrieval tasks.

Feature extraction serves as the core module of cross-modal retrieval, encoding a raw corpus into embeddings, such as vision embedding and language embedding. By applying deep learning models, a sequence of features can be extracted. In contrast to traditional CNN networks [11] that focus on grid features at the pixel level, more recent approaches have emerged that explore region features in images, such as the Faster-RCNN algorithm proposed by [12]. The widespread paradigms of pre-training and fine-tuning have been motivated by the transformer [13] and BERT [14] architectures. For example, ViT [15] can directly process patch features, while BERT, UniLM [16], RoBERTa [17], T5 [18], BART [19], transformer, and ViT support text encoders. For image encoders, there are a variety of options, including Faster-RCNN, ResNet [20], Visual Dictionary [21], Swin transformer [22], EfficientNet [23], and Linear Projection.

Researchers have incorporated pre-training models into cross-modal retrieval systems, modeling the interactions between cross-modal representations. It has been shown that visual concepts in images are critical and complex for cross-modal representations, unlike relationships between words. By extending the BERT model to images and texts, ViLBERT [24] targets region-based object detection and encodes separate sequences of regions using Faster-RCNN. LXMERT [25], similarly to ViLBERT, encodes regions as a sequence of region-of-interest (ROI) features. Apart from region features, pixel-level grid features, such as SOHO [26], CLIP-ViL [27], and pixel-BERT [28], are encoded. They abandon the time-consuming Faster-RCNN. On the contrary, studies are in favor of ResNet extracting grid features. Apart from the region and grid features, patch projection is also used to present image features in many scenarios. ALBEF [29] processes patch features utilizing the ViT encoder directly, generating several flattened 2D patches. OSCAR [30] and ERNIE-ViL [31] develop additional information to facilitate semantic alignments. OSCAR adds region tags as anchor points from images and then implicitly aligns with text words. On the contrary, ERNIE-ViL simulates a scene graph and pays attention to objects with attributions and relations.

The two image- and sentence-retrieval scenarios have been extensively studied to align images and texts with the same semantics [32–37]. At the beginning of a cross-modal alignment study, Reference [32] developed a model that used CNN and Bi-RNN to construct descriptions of pictures and regions. The alignment models incorporate CNN over image regions and bidirectional RNN over sentences. A structured objective leverages a multi-modal embedding to align the two modalities. Carvalho et al. [33] simultaneously leveraged retrieval and class-guided features and formulated a joint objective function and the loss of classification in a shared latent space. The double loss is considered precisely the retrieval loss and class loss. The double triplet scheme brings forward the novel idea of a loss function for cross-modal research. Some researchers presented a dynamic router schema for interactions between different modalities [34]. They designed a framework with four cells to dynamically align fine-grained segments. ViLT [35] utilized linear projections for matching and demonstrated improvements based on alignment pre-training models,

which embed images and captions in the end. ROSITA [36] was motivated by the highlights of OSCAR and ERNIE-ViL and enhanced alignments by integrating cross- and intra-modal knowledge. In addition, another study [37] offered an instance-oriented vision language task architecture that utilized the dot product to align texts and images.

The cross-modal retrieval framework predominantly encompasses fine-grained components: representation, translation, alignment, fusion, and co-learning. This section describes the exact design, comprising essential stages. Figure 2 illustrates the comprehensive architecture of a typical system within this domain. In the full-cycle workflow, these modules are transformed into the following methodologies, including pre-processing, encoder representation, cross-modal attention, and decoder mechanisms. These stages facilitate efficient information extraction and retrieval across different modalities.

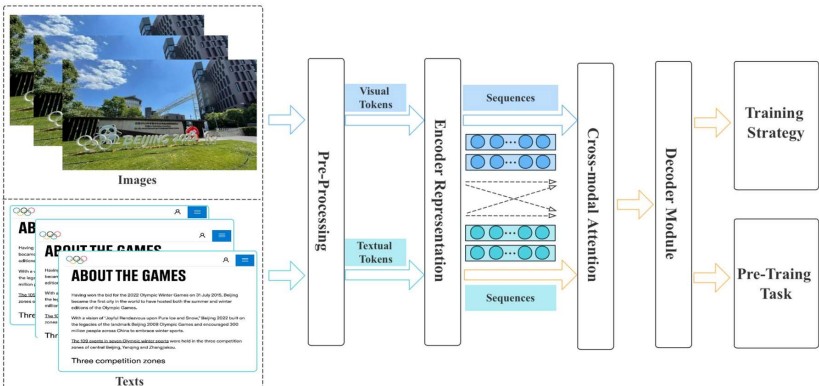

**Figure 2.** An overview of the cross-modal retrieval process.

Pre-processing. The input data are pre-processed to reduce noise and to prepare for subsequent processing. This stage converts image/video and textual phrase inputs into visual and textual tokens. In addition, there are differences between various modalities, so the pre-processing will make a distinction. Apart from the standard tokenization, there are several modules.

Encoder representation. The second period involves representing each modality independently using feature extraction methods. The encoder stage gathers input from visual and textual tokens and generates intermediate states to encode semantic content. After embedding, the most-common methodologies to build an encoder are to utilize LSTM, convolution, and other techniques to encode the token sequences. For text representation, word embeddings, positional embeddings, and segment embeddings are fed into the BERT encoder. Additionally, a series of features, such as image representation, is aligned with a text representation. In this scenario, patch, grid, and region features are extracted from the visual domain.

Vision language pre-training models combine feature extraction and feature fusion with pre-training tasks. These parts address various challenges, such as quantifying the text and image and transmitting them to the model for learning, handling the challenges of representation interactions, and building pre-training tasks to aid models in learning the alignment information. Pre-training on large-scale data can learn semantic correlation across distinct modalities, addressing the issue of difficult access to pricey manual annotations. There are two core pre-training choices with respect to the fusion encoder and dual encoders to aggregate information in the paired data. The single encoder mainly improves the BERT input, whereas double encoders mainly perform co-/cross-BERT. We examined many current publications from 2018 to 2022 and divided them into single-stream and dual-stream models based on how they treat pre-training models. Table 1 shows a road map for pre-training models with fusion encoder and dual encoders. Studies have shown that the single-stream design conducts self-attention on two modalities directly, overlooking intra-modality interaction. As a result, several researchers advocate the adoption of a dual-stream architecture to describe cross-modal interactions.

**Table 1.** A road map table for pre-training models with fusion encoder and dual encoders.

| Dual-Stream Models | Single-Stream Models |
| --- | --- |
| 2019-ViLBERT [24] | 2019-VisualBERT [38] |
| 2019-LXMERT [25] | 2020-UNITER [39] |
| 2020-UNIMO [40] | 2020-Oscar [30] |
| 2020-ViLLA [41] | 2020-Unicoder-VL [42] |
| 2021-ALBEF [29] | 2020-VL-BERT [43] |
| 2021-LightningDot [44] | 2020-E2E-VLP [45] |
| 2021-CLIP [27] | 2020-ImageBERT [46] |
| 2021-ALIGN [47] | 2020-Pixel-BERT [28] |
| 2021-ERNIE-ViL [31] | 2021-ViLT [35] |
| 2019-WenLan1.0(RoBERTa-base) [17] | 2021-VinVL [48] |
| 2022-COTS [49] | 2021-M6 [50] |

A single-stream architecture learns by a single transformer encoder and assumes that it is easy to create a correlation and alignment encompassing vision and language. Furthermore, the features are concatenated with position information before being fed into a transformer-based encoder.

Unlike single-stream architectures, dual-stream architectures utilize a cross-modal mechanism to model two unidirectional cross-attention sublayers. The sublayers are typically composed of a cross-attention layer. They are in charge of transferring information and harmonizing semantics. In this case, parameters are shared between two sublayers, and separate transformers learn contextualized embedding information. As shown in Table 1, single-stream and dual-stream pre-training models have emerged as two prominent types in recent years. A single-stream architecture operates the unordered representation of transformer attention in a unified framework. A number of studies have used the single-stream paradigm for pre-training, such as VisualBERT [24], UNITER [39], OSCAR [30], Unicoder-VL [42], VL-BERT [43], E2E-VLP [45], ImageBERT [46], Pixel-BERT [28], ViLT [35], VinVL [48], and M6 [50]. Some researchers develop segment embedding from different sources to indicate input elements, i.e., VisualBERT and VL-BERT. The dual-stream models include ViLBERT [24], LXMERT [25], UNIMO [40], ViLLA [41], ALBEF [29], LightningDot [44], CLIP [27], ALIGN [47], ERNIE-ViL [31], WenLan1.0 [17], and COTS [49]. In ViLBERT, the co-transformer handles a two-stream interaction. Moreover, the structure has been updated for interactivity, especially considering the text's context while rendering the image. Furthermore, LXMERT is the same as ViLBERT in the pre-training model. UNIMO brings forth new ideas, which take both a single modal and multiple modals into consideration to make a feature fusion. With respect to ViLLA, it employs adversarial training in the pre-training and fine-tuning stages. Adversarial training can help the model generalize better, enabling the performance at the fine-tuning stage. ALBEF presents two categories, yielding strong single-peak and multi-peak representations with enhanced retrieval and reasoning ability. The study of LightningDot proposes to convert costly attention mechanisms into three types of learning objectives.

Cross-modal attention. Much work has been devoted to addressing the representation as mentioned earlier via modeling multi-modal interactions. According to multi-modal representations, correlation modeling is used to learn common representations. The cross-modal interaction encourages other interactions between the two diverse modalities to improve vision language tasks. We classified attention as up-bottom attention, bottom-up attention, recurrent attention, cross-attention, co-attention, distillation-attention, meshed-memory attention, and X-linear attention. The degree of cross-modal information fusion varies among attention mechanisms. Up-bottom attention methods [51] have been widely employed to enable comprehension via fine-grained analysis and even multiple levels

of reasoning. The bottom-up process suggests picture areas, each with its feature vector, while the up-bottom mechanism sets feature weights. According to the study by [52], image text retrieval uses iterated operations and correspondences between visuals and words via repeated alignment stages with recurrent attention memory. This research gains a deeper understanding of fragment correspondences by exploring the attention mechanism. This understanding is compatible with intricate semantics, suggesting using the complex relationship gradually between images and words. Cross-attention conveys encoder and decoder information in [14]. Transformer tracking [53] (TransT) avoids falling into a local optimum of semantic information algorithms. To solve this issue of constructing high-accuracy tracking systems, TransT introduces a unique attention-based feature fusion network. The attention mechanism creates long-distance feature connections, allowing the tracker to focus on important information while extracting a wealth of semantic information. The combination of self-attention and guided attention is known as co-attention. A distillation attention framework [54] is a dual-encoder model that achieves faster inference speeds than a standard fusion encoder thanks to its deep interaction module. In this study, dual-encoder training is guided by fusion encoder instructor information in the annotation, and the proposed knowledge distillation consists of two stages pre-training distillation and fine-tuning distillation, ultimately outperforming other approaches. The use of meshed memory allows the encoder to operate at multiple levels, learning both low-level and high-level relationships. X-linear attention, developed by Pan et al. [55], enables high-order feature interactions, while bi-linear fusion technology improves content interpretation in cross-modal information by capturing second-order interactions between input types using spatial and channel bi-linear attention distributions. Stacked cross-attention is widely used by many researchers to maximize the investigation of vision language features.

Previously, the stack cross-attention network, named SCAN [56], has become a new benchmark for calculating similarity on all potential pairs and not only areas in images and words in sentences. The authors believe sentence descriptions are weakly explainable, implying that words in phrases cannot suit particular positions. As a result, considering the possible connections between visual regions and words, fine-grained interactions between vision and language must be documented. They infer the similarity of entire images and phrases by mapping word and picture areas to a shared embedding space. SCAN uses Faster-RCNN to substitute CNN in DNN and extracts 36 border features with the border indicating the position of candidate regions. Then, the layers of average pooling and fully connection encode outputs are used. In addition, the revolutionary Bi-GRU assesses the relative value of each word in sentences. Bottom-up attention is employed to forecast and encode visual areas to features, and it maps words to sentence context characteristics. Stacking cross-attention infers similarity between visuals and phrases.

Several studies have been derived from the SCAN model, including CAMP [57], IM-RAM [52], MMCA [58], METER [59], and SMAN [60]. For example, CAMP differs from SCAN in that SCAN only interacts in discrete subspaces, whereas CAMP maps image text in the same subspace. CAMP involves specular highlights in vision and salient phrases in language, alternately considers information from the two modalities, and filters out irrelevant information to find fine-grained features for cross-modal matching. CAMP comprises two modules: an aggregate module and a gated fusion module. IMRAM iteratively matches with recurrent attention memory in image text retrieval. Previous studies typically focused on examining all semantic units to ensure uniform alignment, but the complexity of alternative meanings can make it difficult for key semantics to be reflected. To address this issue, iterative matching with repeated attention memory was introduced in IMRAM, which compares the corresponding information between the text and image in several phases to progressively investigate the corresponding fine-grained connections. Given that individuals advance in the retrieval process, a memory distillation unit based on SCAN is introduced to alignment via multi-step iterations. In IMRAM, the image input is for the Faster-RCNN extraction of regional features, followed by a complete connection layer to map each regional feature to a dimensional space. Text input is encoded using bi-GRU to

obtain vector representations of each word in training. By describing the intra-modal and inter-modal interactions between image areas and sentence words in a unified depth model, they presented a novel multi-modal cross-attention (MMCA) network to match images and sentences. In their approach, the authors employed a bottom-up model to extract the characteristics of the significant image region in the self-attention module. Simultaneously, the authors leveraged word token embedding as a linguistic element. The visual domain is then fed into the transformer unit, and the word token is entered into the BERT model to represent the connections between modalities. These features of fragments provide a global representation. The study stacks the representation to picture the representation of regions and sentence words and then passes them. In METER, co-attention and merged attention are two methods for merging trans-modal content. Textual and visual characteristics are passed into separate transformer blocks, adding parameters. Furthermore, textual and visual components are easily integrated into a unique transformer block. Due to global feature alignment, existing approaches cannot distinguish semantic information between images and texts. In contrast, local feature alignment methods have significant computing challenges when aggregating the similarity of vision and language. SMAN offers a stacked multi-modal attention network to investigate fine-grained relationships and aggregates fine fragments into the shared space. Multi-step attention reasoning is accomplished by using modal and multi-modal information as guidance. It is argued that the bi-directional ranking loss encourages a reduction in the distance between pairs of multi-modal examples.

Decoder module. Following the encoding of visual and linguistic feature interaction, the next stage is to use intermediate states to decipher words for every step. Because the decoder module generates outputs in inference, it is the most-comparable to the encoder module. There are various methods for decoding, such as LSTM, GRU, convolution, and transformer. For instance, LSTM outputs each auto-regressive word. Moreover, the transformer first enables word generation via a self-attention and cross-attention mechanism between vision and language. Consequently, the decoder function is opposed to the aforementioned encoder.

## 3. Fine-Grained Deep Learning Methodologies

Fine-grained deep learning approaches focus on advanced feature extraction, learning feature representations, and establishing high-dimensional correlations across various modalities. In this section, we critically review and analyze the full-cycle methodologies employed in the cross-modal retrieval process, highlighting the effectiveness and potential for further improvement.

### 3.1. Feature Engineering

Researchers aim to achieve high-precision retrieval by efficient feature extraction, overcoming complex environments and the network topology. Referring to extensive studies, we classified the feature extraction into global and local features based on the granularity, as shown in Figure 3. Subsequent studies leverage global features, such as VSE++ [61], ACMR [62], and DSPE [63]. In contrast, local features were employed in works such as DAN [64], SCAN [56], SCO [65], and PVSE [66].

We further categorized feature extraction into two types, visual embeddings and textual embeddings, which are crucial components of many cross-modal retrieval systems. Visual embedding greatly influences retrieval efficiency, and current studies are extensive and in-depth. The BERT-like structure is commonly utilized to extract features in textual embedding methodologies. Unlike textual embedding, visual embedding employs different degrees of extraction, including region, gird, and patch levels. The Faster-RCNN, a second-order object detector, is widely used for extracting regional characteristics based on target detection. For example, ViLBERT and LXMBERT employ co-attention to combine multi-modal information. VisualBERT, VL-Bert, and UNITER use merged attention for multi-modal information fusion, whereas OSCAR and VinVL need extra image tags. Despite this, there are significant drawbacks to the approach. Training may freeze object detection.

It limits visual concept recognition and loses context information. Moreover, it cannot describe the connection between many objects. All of the limits described above are based on region extraction characteristics. CNN-based techniques are another popular method of extracting visual features. Grid characteristics are obtained using typical CNN networks in pixel-Bert and CLIP-ViL, whereas text is obtained by using a transformer. SOHO utilizes a learnable visual vocabulary to discretize grid features, which are subsequently fed into multi-modal modules. It performs worse than the OD-based method compared to inconsistent optimizers, i.e., CNN using SGD and transformer using AdamW. The Patch projection enables image slices to extract features. A common approach, such as ALBEF, utilizes ViT directly.

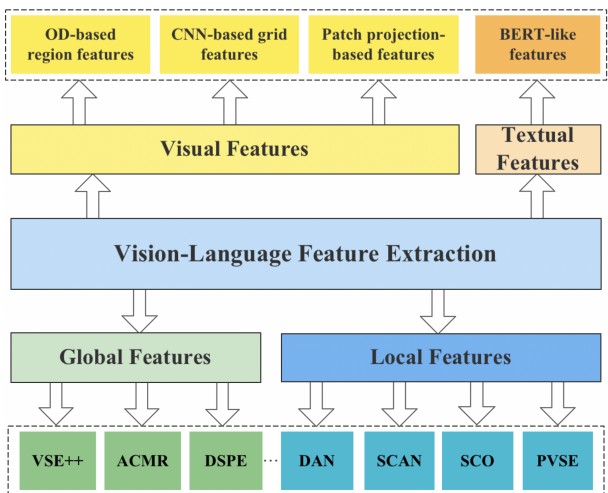

**Figure 3.** Classified diagram of V-L feature extraction.

### 3.2. Cross-Modal Interaction

Compared to the feature representation, the image text matching strategy improves consistency by investigating semantic relationships. Cross-modal interaction plays a critical role in establishing connections between distinct modal representations. This interaction involves matching each pixel, region, or patch to a specific label. There are three primary approaches in cross-modal interaction, namely vision language alignment, vision language reconstruction, and vision language embedding based on semantic associations.

Vision language alignment. Vision language alignment aims to maximize the comparability of image text pairs using large-scale contrastive learning in the dual-encoder model. It employs a re-sharing strategy to address cross-modal heterogeneity between two network branches. In addition, the intra-modal similarity is learned using samples from the exact modal via two conjoined CNN models. In traditional studies, the mode of engagement for cross-modal retrieval has largely relied on manual expert knowledge and empirical input. However, the study [67] proposed a dynamic interaction mechanism for modeling cross-modal retrieval, known as DIME. DIME employs alternative interaction approaches that are tailored to the complexity of the samples. The model includes a local modification cell, an intra-modal inference cell, a global local guidance cell, and a modification cell. ViLT [35] is a novel approach that incorporates visually embedded features via patch projection and patch-level matching of the image and text information. By avoiding the need for time-consuming object identification and convolution techniques with limited expressive capacity, it can effectively improve the performance of cross-modal retrieval. Similarly, in the work by [36], ROSITA employs a pre-training task to enhance fine-grained semantic alignment by suppressing the interference of intra-modal context and eliminating potential noise interference. These advancements demonstrate the effectiveness of these techniques in overcoming the limitations of traditional cross-modal retrieval methods. The ROSITA model draws inspiration from OSCAR and ERNIE-ViL. Additionally, a recent study proposed a new alignment model [68], which embeds images and captions into

the same subspace and enhances image-caption retrieval. The ALBEF model [29] adopts the pre-fusion alignment method and utilizes a transformer-based ViT to collect image features without the use of a CNN. The ViT model employs BERT for text and processes single-modal text using the first six layers and multi-modal processing using the last six layers. The model initially performs self-attention on text, followed by cross-attention and visual feature fusion. Moreover, several studies have extensively explored instancewise alignment. For example, X-DETR [37] introduces a versatile architecture for instance-level alignment and discovers that costly joint modal transformers may be redundant for vision language tasks, while weakly annotated data may be beneficial. X-DETR aligns graphics and texts using the dot product. UVLP [69] demonstrates that the combination of image text alignment and the alignment of the entire image text can achieve excellent unsupervised vision language pre-training without parallel data, based on two critical criteria. The authors proposed constructing a weakly supervised paired corpus and granularity alignment pre-training tasks. Their unsupervised pre-training strategy aims to establish robust joint representations of unaligned texts and images, and the results showed admirable performance across various tasks in an unsupervised setting. The aforementioned alignment approaches have specific criteria pertaining to the dataset size and quality and model granularity, which are essential for achieving optimal results. These techniques emphasize the importance of fine-grained matching in cross-modal retrieval.

Vision language reconstruction. Unlike vision language alignment, the reconstruction focuses more on global information. DSPE [63] learns from image text embedding to solve matching problems. The optimization of the loss function aims to improve the distribution of features in high-dimensional space, resulting in a more effective clustering effect. MASLN [70] proposes a solution to the issue of classes being unable to traverse instances. The proposed solution involves the use of a reconstruction sub-network, which rebuilds each modal dataset using conditional autoencoders. The sub-network leverages information from the input to the output, while minimizing discrepancies in the distribution. Additionally, MASLN introduces an adversarial sub-network to develop semantic representations. The referenced study [71] investigated neural networks for embedding and similarity calculations. The embedding network learns a latent embedding space with a new neighborhood restriction and a maximum margin ranking error. The authors improved neighborhood sampling to produce tiny batches compared to ordinary triplet sampling. The similarity network uses an elementwise product and applies regression loss training to forecast the similarity score directly. A significant number of trials indicate that this network can accurately locate phrases. The problem of visual and textual retrieval has been reformulated as a text and vision conversion task in recent research [72]. To address this task, a cycle-consistent network was proposed by the authors. In another related study [73], the attention mechanism was enhanced by incorporating a scene graph structure. Specifically, the sentence reconstruction network creates a scene graph from the objects, attributes, and relations extracted by the detection network. The resulting graph is then processed by a graph convolution network to generate a word vector, which is subsequently fed into a pre-trained dictionary shared by the encoder decoder model. This approach results in more natural and human-like visual descriptions in the generated corpus.

The study on reconstruction overcomes the constraint of embedding spaces. The reconstruction approach employs a deep autoencoder to minimize heterogeneity and improve the semantic discrimination capacity. Furthermore, compared to cross-modal alignments, cross-modal reconstruction has fewer dataset requirements and a lower annotation cost, making it suited for small- and medium-sized datasets.

Vision language embedding. Joint embedding may develop superior feature discrimination by integrating global and local information as semantic feature embedding. The study of DSCMR [74] presented a supervised learning structure to retain semantic distinction and modal invariance. It creates two sub-networks with weight-sharing restrictions. The authors reduced discrimination losses in labels and common representation space, increasing the significance of the learned common representation. The learning strat-

egy of DSCMR can fully unitize paired labels and classification information, successfully learning the typical representation of heterogeneous data. PCME [75] matches one image to numerous titles or one title corresponding to multiple images. The authors believe that most existing models' deterministic functions are insufficient to capture a one-to-many correspondence. A federated representation space PCME paradigm maps one-to-many relationships. It uses probabilistic maps and does not need the precise formulation of many-to-many matches. The uncertain estimation enables PCME to evaluate retrieval difficulty and failure probability, i.e., the auxiliary interpretability aspect. The probabilistic model learns from a more decadent embedding space, where set relations are also beneficial, whereas only similarity relations are helpful in precise spaces. Probability mapping is a supplement to the accurate retrieval system. ViSTA [76] presents a transformer framework for learning an aggregated visual representation by directly encoding patches and scene embedding. It proposes a novel aggregation token to embed pairs and combine them into the shared space. The bidirectional contrastive learning loss tackles the modal loss problem of the scene text.

This joint embedding strategy focuses on high-level semantics. Rich semantic correlation approaches can successfully address the polysemy instance. Moreover, vision language embedding can enhance the accuracy and expansibility of image text matching. In addition, the embedding has a strong retrieval performance.

### 3.3. Pre-Training Tasks

The input is unstructured in cross-modal retrieval and transformed into the format of vectors. From previous studies, the data-driven pre-training models may learn from it and is highly impacted by the results of pre-training tasks. We classified and summarized pre-training tasks in cross-modal retrieval and divided them into text-based, vision-based, and cross-modal tasks. A glossary of pre-training tasks is outlined in Table 2. We show how to use pre-training tasks to train models, which are critical for a universal representation. Pre-training tasks' primary objectives include sequence completion, pattern matching, and providing temporal/contextual features.

**Table 2.** Glossary of pre-training tasks.

| Pre-Training Type | Task Name |
|---|---|
| Vision-Based Tasks | MOC: masked object classification<br>MRFR: masked region feature regression<br>MRM: masked region modeling<br>MFR: masked feature regression<br>MFC: masked feature classification<br>MRC: masked region classification<br>MIM: masked image modeling |
| Text-Based Tasks | MLM: masked language modeling<br>NSP: next sentence prediction<br>WRA: word region alignment<br>PLM: permuted language modeling<br>CLTR: cross-lingual text recovery<br>TLM: translation language modeling |
| Cross-Modal Tasks | VLM: visual-linguistic matching<br>ITM: image text matching<br>MTL: multi-task learning<br>CMCL: cross-modal contrastive Learning<br>CMTR: cross-modal text recovery<br>PrefixLM: prefix language modeling<br>DAE: denoising autoencoding<br>ITCL: image-text contrastive learning<br>MRTM: masked region-to-token modeling<br>VTLM: visual translation language modeling |

### 3.4. Unified V-L Architecture

This section introduces how we study the unified architecture, which is crucial for learning vision and language information. We summarize the vision language (V-L) architecture into two categories: the universal representation and unified generation model from recent references. The universal representation aims to learn a single embedding space whereby multiple modalities may be represented. A unified generation model is a form of cross-modal retrieval that utilizes a single model to build content representations in several modalities. Both approaches have their advantages and disadvantages, and the choice of approaches depends on the specific requirements. Firstly, we present an overview of the two architectures in this section. Subsequently, we provide a comprehensive evaluation of the advantages and disadvantages, highlighting their strengths and weaknesses.

Universal representation. Universal representation is essential for effectively comparing similarity across different modalities in cross-modal retrieval. To achieve this goal, the DSCMR model proposed by [74] presents a generic representation space that enables the direct comparison of samples from multiple modalities. The framework employs a supervised cross-modal learning approach to establish connections between disparate modalities, successfully learning common sentences while retaining semantic distinctions and modal invariance. To discover cross-modal correlations, the final layer of the model contains two subnets with weight-sharing restrictions. Modal invariance losses are incorporated into the objective function to remove discrepancies, and a linear classifier categorizes the data in the common representation space. These features collectively make the DSCMR model a promising approach for cross-modal retrieval. The proposed method in SDML [77] defines a public space beforehand while simultaneously maximizing the smallest group gap. SDML is the first model to support an infinite number of modal inputs. To train a specific network for different modalities, the input is projected into a predefined subspace. This approach trains additional modalities without learning all modals simultaneously. UNITER aims to solve the problem of determining whether it learns a common vision language representation for all V-L tasks. Its large-scale pre-training process allows it to handle diverse downstream V-L tasks and joint multi-modal embedding.

Besides the joint representation, the universal encoder has also been studied extensively. For example, Unicoder-VL develops a generic vision and language encoder. Unicoder-VL employs three types of pre-training tasks, including MLM, MOC, and VLM. The tasks collaborate to create context-aware representations of input tokens. It also tries to predict whether a picture and a text are related and performs other algorithms without jointly pre-training for image-text retrieval. It illustrates that transferring learning may also produce excellent results in cross-modal tasks. GPV [78] provides a general purpose and task-agnostic system. It receives visual characteristics and textual descriptions. Besides, it generates bounding boxes, confidences, and output information. Without affecting the network structure, the system may learn and carry out any task across a large domain. GPV comprises an optical encoder, a textual encoder, and a co-attention module. The CNN backbone and the DETR transformer encoder–decoder are used to create an object detector. It also refers to ViLBERT, which can encode cross-contextualize representations from visual and linguistic encoders. Because collecting and annotating task-specific data in all languages is unfeasible, there is a strong need for a framework to make universal models across languages. M3P [79] provides a multi-language and multi-modal pre-training paradigm that integrates them into a cohesive framework to acquire universal representations. It leverages the inadequate supervision of multi-lingual text video data, inspired by recent achievements in large-scale language modeling and multi-modal pre-training.

Unified generation model. The discriminative model and generative model may be classed. Several works have investigated a general framework from the standpoint of model development. Due to the growth of cross-modal retrieval, the single task framework cannot meet the needs of multiple tasks. Therefore, the study [80] explored a unified framework based on a text generation model. The framework is simultaneously compatible with multi-modal task learning. The approach is conditional text generation, which means that images

and texts produce text labels, and the knowledge between tasks may be shared. Moreover, UNICORN [81] bridges the texts and boundaries of box formats, aiming at unified vision language modeling. Text generation and bounding box prediction are combined in this model framework, which can dynamically design different heads for various problems. The Pix2Seq model is a general-purpose target detection framework inspiring UNICORN. A discrete approach is employed to convert the bounding box location into a discrete token sequence. Generative adversarial networks improve the synthesis of images by learning the underlying data distribution. However, there has seldom been research on other visual tasks using image-generating tasks. VILLA is the first technique integrating large-scale adversarial training to boost model generalization. It is a comprehensive framework that utilizes any pre-training model to increase the model's generalization capacity. To put it another way, VILLA employs confrontational learning in the stages of pre-training and fine-tuning. As a branch of self-supervised learning technology in deep learning, the unified generation model focuses on defining the data production process.

The pros and cons of the V-L architecture are summarized in Table 3. A universal representation offers several advantages, such as improved accuracy, better generalization, and increased efficiency, by reducing the computational resources and training time for multiple tasks. However, it also presents challenges in terms of increased complexity, potential loss of modality-specific information, and limited interpretability due to the intricacy of interactions between vision and language. On the other hand, unified generation models possess the ability to generate outputs in one modality based on inputs from another, resulting in better performances in cross-modal retrieval. Nonetheless, these models exhibit limited flexibility, increased complexity during training, and a higher risk of overfitting, primarily because they generate representations for multiple modalities simultaneously, which may require diverse training data to prevent overfitting.

**Table 3.** Pros and cons of V-L architecture.

| | |
|---|---|
| Pros of Universal Representation | **Improved Accuracy** : It reduces the computational resources and training time needed, making retrieval faster and more efficient. |
| Pros of Universal Representation | **Better Generalization**: A universal representation can lead to better generalization. This can improve performance and reduce the need for large amounts of training data. |
| Pros of Universal Representation | **Increased Efficiency**: A universal representation reduces the computational cost of developing and training models. Instead of creating separate models for each task, the model can be trained and used for multiple tasks. |
| Cons of Universal Representation | **Increased Complexity**: A universal representation is a complex and challenging task that requires a significant cost of time and resources. Developing cross-modal retrieval model may require expertise from multiple domains and may involve complex algorithms and architectures. |
| Cons of Universal Representation | **Loss of Modality-Specific Information**: Combining multiple modalities into a single model, some modality-specific information may be lost. This may reduce the accuracy of the cross-modal retrieval that require fine-grained features. |
| Cons of Universal Representation | **Limited Interpretability**: A universal representation may be difficult to interpret, making it challenging to understand the interactions between vision and language. This lack of interpretability may be a concern for applications. |

**Table 3.** *Cont.*

| | |
|---|---|
| Pros of Unified Generation Model | **Generation Ability**: A unified generation model can generate outputs in one modality based on inputs from another modality, which can be useful for cross-modal retrieval. |
| Pros of Unified Generation Model | **Better Performance**: A unified generation model can provide better performance in cross-modal retrieval compared to separate models for different modalities. The unified model can capture the complex relationships between different modalities more effectively. |
| Cons of Unified Generation Model | **Limited Flexibility**: A unified generation model is not as flexible as traditional models in handling different modalities. The model generates representations for all modalities, which may not be optimal for specific modalities. |
| Cons of Unified Generation Model | **Increased Complexity of Training**: Although training a unified generation model may take less time, the complexity of training the model may be higher. The model generates representations for multiple modalities, which can be a more challenging task. |
| Cons of Unified Generation Model | **Increased Risk of Overfitting**: A unified generation model is more prone to overfitting. The model generates representations for multiple modalities simultaneously, which may overfit if the training data are not sufficiently diverse. |

## 4. Loss Function

The loss function will assess the model's performance by comparing the anticipated and expected outputs of the model and then determining the directions of optimization. If the difference between the two is exceptionally high, the loss value will be significant. Oppositely, if the difference is tiny or about equal, the loss value will be meager. As a result, an adequate loss function is required that correctly punishes the model as it is trained on the dataset. This section defines the principal loss function and performance analysis methods. We summarize innovative samples of the loss function in cross-modal tasks, which can be seen in Figure 4.

Conventional classification methodologies divide loss functions into regression, binary, and multi-class categories. In contrast, our survey synthesized the utilization and development of loss functions within the context of cross-modal retrieval, ultimately distilling the loss function into four fundamental components. Consequently, the loss functions were classified into regression-based loss, classification-based loss, ranking loss, and cross-modal application loss. For example, the L1 loss and L2 loss represent prevalent approaches for calculating loss in regression loss functions. The L1 loss function calculates the mean absolute difference between predicted and actual values, with a valid domain extending from zero to positive infinity. This loss function exhibits rapid convergence, and the gradient can be assigned an appropriate penalty weight rather than an equal one, thus providing a more accurate gradient for updating the optimization direction. However, the L1 loss is vulnerable to the influence of outliers, which can dominate the gradient update process and render it unreliable. In contrast, the L2 loss function computes the sum of squared differences between predicted and true values. Despite its merits, the L2 loss presents certain drawbacks, such as a discontinuous derivative at 0, which leads to reduced solution efficiency and sluggish convergence. Additionally, the gradient for small loss values is indistinguishable from that of other loss values, which is not conducive to effective network learning. In the context of classification-based loss functions, the cross-entropy loss function is predominantly utilized. Ranking loss, on the other hand, estimates the relative distance between the input samples and is often associated with metric learning. Triplet

loss serves as the most-prevalent method for assessing matching models, with random samples, positive examples, and negative examples being essential components to consider.

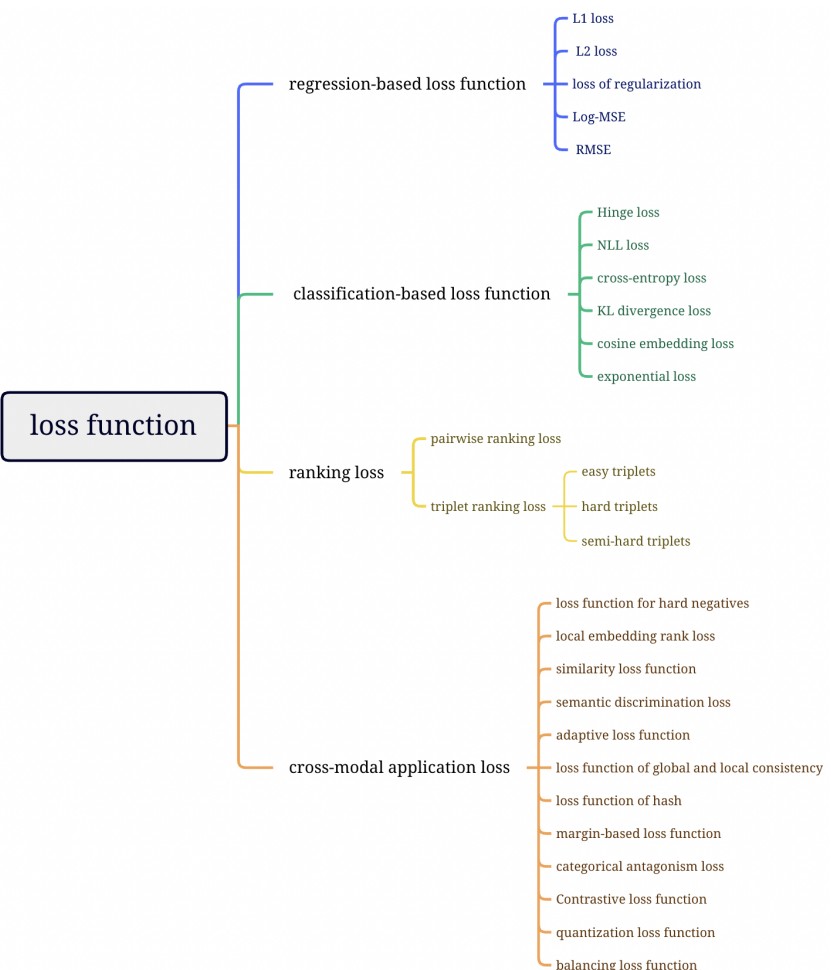

**Figure 4.** Innovative samples of the loss function.

In recent applications of cross-modal retrieval, specific loss functions have been adeptly integrated into model training procedures. In the study conducted on VSE++ [61], hard negative samples were employed to enhance the retrieval methodology in a visual semantic embedding framework. The researchers opted for max of hinges (MH) loss rather than sum of hinges (SH) loss. SH loss accounts for the aggregate of all values, whereas MH loss selects the maximum value among all triplet losses. The modification of conventional loss function applied to multi-modal embedding was motivated by the exploration of hard negatives, the incorporation of hard negative samples in structural prediction, and the rank-based loss function. When combined with fine-tuning strategies and the implementation of augmented data, this approach significantly elevates retrieval performance. In the study of [82], an innovative approach employing adaptive loss was applied to the realm of fashion, facilitating cross-modal retrieval by enabling searches using either text or images. By integrating textual and visual elements into the transformer architecture, adaptive loss was employed to modify the weightings of text matching and image matching. Reference [83] generated images in a feature embedding framework. They proposed a dual-structure embedding that consists of a global semantic embedding and a supplementary local embedding to enhance the performance. The ranking loss is a key component in their approach, with the bidirectional autoencoder effectively learning features in public spaces. Furthermore, they demonstrated that ranking loss with a violation penalty outperforms hard triplet loss in the context of VSE++. A number of scholars employ a diverse range of loss functions in their

research. These functions may include global consistency loss within a textual modality, local and global loss across modalities, and categorical antagonism loss. Some researchers constructed hybrid networks and precisely defined loss functions, as outlined by [84]. Such functions encompass hash loss, similarity loss, quantization loss, and equalization loss. The aforementioned network facilitates the concurrent training of deep representations and quantizers. The study [85] introduced a comprehensive approach incorporating multiple loss functions to enhance the model's efficacy. These functions encompass semantic differentiation loss, contrastive loss, and significant margin loss. Reference [63] introduced a soft margin triplet loss for noise data processing, addressing noise correspondence. A new polynomial loss function was developed in [86] to determine the polynomial weight with positive and negative samples. This function is capable of selecting informative pairs from redundant pairs. Moreover, the loss functions were discussed separately in [87], and the learning strategy was divided into four parts. With respect to mitigating the semantic discrepancy, the authors introduced a discriminative label function, shown in (Equation (1)):

$$\ell_{label} = \frac{1}{N}\left(\left\|\varphi^T F_{I_1}, F_{I_2}, \ldots, F_{I_N}\right\|_F + \left\|\varphi^T F_{T_1}, F_{T_2}, \ldots, F_{T_N}\right\|_F\right) \tag{1}$$

where $F_{I_i}$ represents the *i*-th image in common subspace and $F_{T_i}$ is the *i*-th text's representation. $N$ denotes the number of image–text pairs. The notation $\varphi$ is for classification as the parameter vector. $T$ is the text sample in a multimedia database. The Frobenius norm is defined as $\|\cdot\|_F$. Classifier $(F_{I_i}; \varphi)$ and Classifier $(F_{T_i}; \varphi)$ are two classifiers to recognize the semantic discrimination in label space. In the study of [87], the authors preserved visual and textual representations in a unified subspace. The objective loss function encompasses three dimensions:

$$
\begin{aligned}
\ell_{common} = \frac{1}{N^2}\Bigg( & \sum_{i,j=1}^{N}\left(\log\left(1+e^{\frac{1}{2}\cos\left(F_{I_i},F_{T_j}\right)}\right) - \frac{1}{2}\cos\left(F_{I_i},F_{T_j}\right) \equiv \left(F_{I_i},F_{T_j}\right)\right) \\
& + \sum_{i,j=1}^{N}\left(\log\left(1+e^{\frac{1}{2}\cos\left(F_{I_i},F_{I_j}\right)}\right) - \frac{1}{2}\cos\left(F_{I_i},F_{I_j}\right) \equiv \left(F_{I_i},F_{I_j}\right)\right) \\
& + \sum_{i,j=1}^{N}\left(\log\left(1+e^{\frac{1}{2}\cos\left(F_{T_i},F_{T_j}\right)}\right) - \frac{1}{2}\cos\left(F_{T_i},F_{T_j}\right) \equiv \left(F_{T_i},F_{T_j}\right)\right)
\end{aligned}
\tag{2}
$$

In the above Equation (2)), cross-modality, image, and text semantic discrimination loss are learned, where function $\equiv (,)$ represents an indicator function; if both inputs are from the same modality, this function has a value of 1; else, it is 0.

Cross-modal representations can learn modality invariance to minimize heterogeneity across distinct modalities by decreasing the distance in the shared subspace. The objective function of invariance loss is as follows:

$$\ell_{invar} = \frac{1}{N}\|(F_{I_1}, F_{I_2}, \ldots, F_{I_N}) - (F_{T_1}, F_{T_2}, \ldots, F_{T_N}) \tag{3}$$

Researchers investigate semantic associations by combining the similarity of inter-modal and intra-modal. The $S$ denotes the feature vector of the image generated by Siamese CNNs. $L$ is the classification label of the i-th image-text pair. The intra-modal similarities are paired. $\langle \mathbf{I}_i, \mathbf{T}_i, \mathbf{L}_i \rangle$, $\langle \mathbf{I}_j, \mathbf{T}_j, \mathbf{L}_j \rangle$ and $\mathbf{L}_i = \mathbf{L}_j$, while the cross-modal similarities are $\text{Sim}(\mathbf{I_i}, \mathbf{T_i})$ and $\text{Sim}(\mathbf{I_j}, \mathbf{T_j})$. The hybrid loss is:

$$
\begin{aligned}
\ell_{hybrid} = & \left| \text{Sim}\left(\mathbf{F}_{I_i}, \mathbf{F}_{T_j}\right) - \text{Sim}\left(\mathbf{S}_{I_i}, \mathbf{S}_{I_j}\right) \text{Sim}\left(\mathbf{F}_{I_j}, \mathbf{F}_{T_j}\right) \right. \\
& \left. - \left( \text{Sim}\left(\mathbf{S}_{I_i}, \mathbf{S}_{I_j}\right) - \text{Sim}\left(\mathbf{F}_{I_i}, \mathbf{F}_{T_i}\right) \text{Sim}\left(\mathbf{F}_{I_j}, \mathbf{F}_{T_i}\right) \right) \right|
\end{aligned}
\tag{4}
$$

To avoid gradient exploding or vanishing problems, researchers train models by employing various loss functions for relationship prediction and object classification. Discriminative characteristics can be monitored via the loss function, which minimizes discrim-

ination loss in both common and label spaces. The binary encoding of hash representation frequently leads to information loss, leading to a decline in accuracy. To address this concern, researchers have introduced real-valued representation learning methodologies to enhance the overall accuracy. Moreover, these studies facilitate direct optimization of the complete pipeline through multi-level rewards.

## 5. Evaluation Metrics

There are a variety of evaluation indicators to demonstrate the efficacy of cross-modal retrieval. The effectiveness of a methodology in a particular scenario is evaluated using appropriate metrics [88]. In this section, we make a comparison with predominant evaluation metrics such as the precision *(P)*, recall rate *(Recall@K)*, PR curve *(PR)*, mean average precision *(mAP)*, F-score *(FS)*, and normalized discounted cumulative gain *(NDCG)*.

Precision. Precision is the ratio of accurately recovered samples to total retrieved samples in cross-modal systems.

Recall rate. The recall rate is an important performance metric in cross-modal retrieval, measured by *recall@k*. *Recall@1*, *@5*, and *@10* represent the recall rate of the first k retrieved results. To evaluate the effectiveness of a retrieval system in specific scenarios, the *Recall@50* and *Recall@100* metrics are often used. It should be noted that achieving a balance between recall and accuracy is crucial since they are inversely related. Equation (5)) represents the calculation of the recall metric.

$$Recall = TP/(TP + FN) \tag{5}$$

where *TP* stands for the total number of documents returned from the retrieval that matches the query and *FP* represents the number of unmatched samples in the retrieved samples. The variable *FN* denotes the number of documents that fail to match the query sample and are not returned from the dataset.

PR curve. The performance of cross-modal retrieval can be evaluated by combining recall and precision measures to produce a precision–recall curve *(PR curve)*. This curve displays the precision value at various recall levels and provides a visual representation. For example, Reference [89] utilized the *PR curve* to demonstrate the effectiveness of their proposed cross-modal retrieval method.

mAP. The *mAP* metric evaluates the relevance of the retrieved results to the query and is calculated as the average accuracy of all queries. The studies cited in [74,87,90] employed the *mAP* metric to improve retrieval performance. The average accuracy of a query and the top-K retrieved results can be computed using Equation (6):

$$AP = \frac{1}{R} \sum_{r=1}^{R} P(r)\delta(r) \tag{6}$$

where *P(r)* signifies the precision of the top *R* retrieved outputs and *R* is the number of relevant outcomes, and if the retrieved *R* is relevant, the value is set to 1; otherwise, it is set to 0. The mean average precision *(mAP)* can be specified as (Equation (7)):

$$mAP = \frac{1}{Q} \sum_{q=1}^{Q} AP \tag{7}$$

A high *mAP* value indicates that a specific cross-modal method performs well on a given dataset, based on a certain number of queries *(Q)*. Unlike the *precision*, *recall*, and *F-score*, which are single-point values, the *mAP* provides a comprehensive measure of retrieval capabilities and represents overall performance.

F-score. The *F-score*, which is the weighted harmonic mean of the *precision* and *recall*, is a widely used performance metric in various fields. It is calculated using the following formula (Equation (8)):

$$F = \left(1 + \beta^2\right) * \frac{precision * recall}{\beta^2 * (precision + recall)} \tag{8}$$

One important factor in measuring performance is the weight given to precision and recall. This weight can be adjusted using the $\beta$ parameter, where a value of 1 corresponds to equal weighting of precision and recall, known as the *F1-score*. The $\beta$ parameter can tune the relative importance of precision versus recall in the retrieval process. As a result, the *FS-score* is a popular metric for evaluating retrieval performance, as it provides a balanced optimization of the precision and recall values. For example, in the study of [91], the *FS-score* was used to measure the performance of the retrieval system.

NDCG. Normalized discounted cumulative gain (*NDCG*) is a widely used method for evaluating search engine ranking performance. It assesses the quality of sorted recommended items, typically with a list length of *L*. A higher correlation between the sorted list and the actual preferences results in a higher *NDCG* score. Specifically, the impact on the final *NDCG* score is greater if it appears in a higher position and has a higher correlation with user preferences. Additionally, memory requirements and retrieval computing speed are also considered with the development of deep learning models and more corpus to process.

## 6. Benchmark Datasets

Benchmark datasets are commonly utilized to evaluate the performance of cross-modal retrieval. Table 4 displays the analysis and explanation of classical cross-modal datasets, including the names of the datasets, the numbers of images and texts, and descriptions.

**Table 4.** Summary of representative datasets facilitating cross-modal retrieval.

| Name | Number | Description |
|---|---|---|
| NUS-WIDE [92] | 269,648 | Every image includes 2 to 5 label claims on average. |
| https://www.image-net.org/ (accessed on 1 January 2022) | 14,197,122 | ImageNet aims at classification, positioning, and detection task evaluation. |
| Pascal VOC 2007 [93] | 9963 | It includes training, validation, and test and marks 24,640 objects. |
| Pascal VOC 2012 [94] | 11,530 | Each image is tagged with 20 categories of objects, including people, animals, vehicles, and furniture. Each image has an average of 2.4 objects. |
| Wikipedia [95] | 2866 | The most-often used dataset for retrieval study is Wikipedia, which consists of entries with relevant picture–text pairs. |
| SBU Caption [96] | 1,000,000 | Image captions are a retrieval task containing 1 million image URLs and title pairs. Humans write the captions, which are then filtered to leave only those with at least two nouns, noun–verb pairs, or verb–adjective pairs. |
| Flickr 8k [97] | 8092 | Each image is accompanied by five human-generated subtitles that focus on humans or animals performing an action. |
| Wiki-CMR [98] | 74,961 | A web cross-modality dataset includes written paragraphs, photos, and hyperlinks. |

**Table 4.** *Cont.*

| Name | Number | Description |
|---|---|---|
| Flickr 30k [99] | 31,783 | The images are accompanied by 1,58,915 captions gathered through crowdsourcing. |
| MS-COCO [100] | 3,28,000 | The dataset contains 25,00,000 annotated occurrences. The collection contains artifacts from 91 different categories. |
| Visual Genome [101] | 108,077 | There are about 5.4 million captions given to picture areas. It contains around 2.8 million labels for object properties in the picture and approximately 2.3 million labels for object connections. |
| PKU FG-XMedia [102] | 50,000 | Over 50,000 samples are included in the PKU FG-XMedia collection, comprising 11,788 graphics, 8000 texts, 18,350 videos, and 12,000 audios. It has different media kinds, a precise granularity of categories, and multiple data sources. |
| Conceptual Caption [103] | 3,000,000 | The photographs and descriptions are gathered from the Internet, and they feature a diverse spectrum of emotions. The captions are derived from the HTML alt element of each picture. |
| Objects 365 [104] | 630,000 | The collection includes 630,000 photos from 365 categories and up to 10 million frames. It is distinguished by its cast scale, excellent quality, and good generalizability. |
| https://storage.googleapis.com/openimages/web/index.html (accessed on 1 January 2022) | 9,000,000 | Open Images is a library of 9 million collection including image-level labels, 15,851,536 bounding boxes, 2,785,498 segmentation masks, visual connections, 675,155 localized narratives, and 59,919,574 image-level labels. |
| M6 [50] | 100,000,000 | It is the most-extensive dataset of a Chinese multi-modal pre-training model, with 1.9 T pictures and 292 G texts. |
| Conceptual 12M [105] | 12,000,000 | It includes a much greater spectrum of visual concepts than the Conceptual Captions dataset, used for image captioning model pre-training and end-to-end training. |

### 6.1. Cross-Modal Datasets

Benchmark datasets are vital for evaluating the performance of cross-modal retrieval. They enable researchers to compare and contrast different techniques and algorithms under consistent conditions, providing a comprehensive and accurate assessment of their strengths and limitations. A high-quality dataset that includes multi-modal data, high-quality labels, and diverse content is crucial for training and evaluating models and advancing relevant research. Nonetheless, conventional datasets might be tailored for a particular task, thereby constraining their versatility in addressing different downstream tasks. The high-quality datasets should encompass a broad range of contextual information and scenarios to ensure the model can learn from multiple domains and exhibit robustness in cross-modal retrieval. Researchers create multi-modal datasets to evaluate the efficacy of the proposed cross-modal methodologies. There are some publicly available datasets in cross-modal retrieval that have been obtained through large-scale collection and labeled.

For example, the NUS-WIDE [92] dataset is frequently used in cross-modal hashing. Researchers can investigate the difficulties of image annotation and retrieval research based on NUS-WIDE. ImageNet (https://www.image-net.org/, accessed on 1 January 2022) is for items in an image to have multiple appearances, positions, views, postures, background clutter, and occlusion. MS-COCO [100] contains more images in each category than the ImageNet dataset, allowing for more scenarios to obtain. The MS-COCO dataset only annotates 80 object categories and does not describe all objects in the image. In the traditional scenario, there might be more object categories. The Visual Genome [101] dataset labels all visual objects in the images with the Objects Categories. Wikipedia [95] is the most-extensively used for cross-modal retrieval. However, it contains limited samples and semantic categories based on Wikipedia articles. SBU Captions [96] contains images with one-million user-generated titles. However, these pairs are still insufficient to train models with hundreds of millions of parameters. The Flickr 8K [97] and Flickr 30k [99] datasets come from Flickr, Yahoo's photo album site. The images in both datasets typically depict individuals engaged in an activity. The handwritten annotation for each image is five words. Because the two databases derive from the same root, the syntax of the annotations is comparable. Besides, twelve million image–text data pairs named Conceptual 12M [105] are suitable for training vision-and-language models. The researchers explore and compare this dataset to the previously popular dataset. The new dataset emphasizes long-tail visual identification, and the quantitative and qualitative findings indicate the benefits for visual and language tasks. Data from multiple media types are distributed and represented differently in fine-grained cross-media retrieval. Some researchers created a fine-grained cross-media retrieval dataset called PKU FG-XMedia. Moreover, the Object365 [104] dataset, which contains objects in natural situations,primarily handles large-scale detection problems, including 365 object types, and it provides a diverse and useful baseline for target detection research. Open Images V6 (https://storage.googleapis.com/openimages/web/index.html, accessed on 1 January 2022) significantly increases the annotation of the Open Images dataset when compared to V5, introducing many new visual associations, human activity annotations, and horizontal picture labels. Some academics focus on creating large-scale Chinese datasets to undertake large-scale multi-modal pre-training in Chinese, such as M6 [50]. The M6 dataset aids large-scale pre-training models in learning sophisticated global information in Chinese, including sports, politics, science, and other disciplines.

Researchers can leverage benchmark datasets to develop more effective and efficient cross-modal retrieval systems, resulting in advancements in image retrieval, text retrieval, and cross-modal retrieval, among other applications.

### 6.2. Comparison on Flickr 30k and MS-COCO Datasets

The Flickr 30k and MS-COCO datasets are widely employed for evaluating cross-modal representations. Our comparisons relied on the test sets from Flickr 30k and MS-COCO. We utilized metrics such as IR@K and TR@K (K = 1, 5, 10), where R_SUM represents the total recall. We present the comparison results on the Flickr 30k dataset in Table 5. Non-pre-trained models were trained from scratch. Generally, models incorporating pre-training exhibit superior performance compared to traditional methods without pre-training. For instance, SCAN [56] maps the features of each modality to a low-dimensional space, employing only shallow interactions for similarity calculation. CAAN [106] suggests collecting local information and investigating intra-modal properties. This approach overlooks significant cross-modal commonalities. While VSRN [107], MMCA [58], and DIME [108] utilize distinct methods to employ semantic reasoning or dynamic interaction, they need to gain early learning of common properties across modalities. Therefore, the effect of feature training is inferior to the result of the pre-training mode.

**Table 5.** Cross-modal retrieval performance comparison on Flickr 30k (1K test set).

| Models | Pretrain Images | Flickr 30k (1K Test Set) | | | | | | R_SUM |
|---|---|---|---|---|---|---|---|---|
| | | TR@1 | TR@5 | TR@10 | IR@1 | IR@5 | IR@10 | |
| SCAN | - | 67.4 | 90.3 | 95.6 | 48.5 | 77.7 | 85.2 | 464.7 |
| CAAN | - | 70.1 | 91.6 | 97.2 | 52.8 | 79.0 | 97.9 | 488.6 |
| VSRN | - | 71.3 | 90.6 | 96.0 | 54.7 | 81.8 | 88.2 | 482.6 |
| MMCA | - | 74.2 | 92.8 | 96.4 | 54.8 | 81.4 | 87.8 | 487.4 |
| CAMRA | - | 78.0 | 95.1 | 97.9 | 60.3 | 85.9 | 91.7 | 508.9 |
| DIME | - | 81.0 | 95.9 | 98.4 | 63.6 | 88.1 | 93.0 | 520.0 |
| UNITER | 4 M | 87.3 | 98.0 | 99.2 | 75.6 | 94.1 | 96.8 | 551.0 |
| VILLA | 4 M | 87.9 | 97.5 | 98.8 | 76.3 | 94.8 | 96.5 | 552.8 |
| ALIGN | 1.8 B | 95.3 | 99.8 | 100 | 84.9 | 97.4 | 98.6 | 576 |
| UNIMO | 5.7 M | 89.4 | 98.9 | 99.8 | 78 | 94.2 | 97.1 | 557.4 |
| VLC | 5.6 M | 89.2 | 99.2 | 99.8 | 72.4 | 93.4 | 96.5 | 550.5 |

Table 6 presents the comparison results on the MS-COCO test dataset. These findings suggest that the larger the pre-training dataset, the higher the recall in cross-modal retrieval is. ALIGN [29], which utilizes 1.8B pre-training data, outperforms algorithms with the 4M and 5.6M pre-training data in retrieval tasks. UNITER [39] and ViLT [35] employ single-stream encoders. However, since UNITER relies on a pre-trained object detection model, it introduces additional noise, leading to suboptimal retrieval performance. In contrast, ViLT incorporates an RPN-like candidate region generator, but the limited tags in supervised training pose challenges, resulting in performance constraints.

**Table 6.** Cross-modal retrieval performance comparison on MS-COCO (5K test set).

| Models | Pretrain Images | MS-COCO (5K Test Set) | | | | | | R_SUM |
|---|---|---|---|---|---|---|---|---|
| | | TR@1 | TR@5 | TR@10 | IR@1 | IR@5 | IR@10 | |
| UNITER | 4 M | 65.70 | 88.60 | 93.80 | 52.90 | 79.90 | 88.00 | 468.90 |
| ViLT | 4 M | 61.50 | 86.30 | 92.70 | 42.70 | 72.90 | 83.10 | 439.20 |
| ALBEF | 4 M | 73.10 | 91.40 | 96.00 | 56.8 | 81.50 | 89.20 | 488.00 |
| TCL | 4 M | 75.60 | 92.80 | 96.70 | 59 | 83.20 | 89.90 | 497.20 |
| VLC | 5.6 M | 71.30 | 91.20 | 95.80 | 50.70 | 78.90 | 88.00 | 475.90 |
| ALIGN | 1.8 B | 77.00 | 93.50 | 96.90 | 59.90 | 83.30 | 89.80 | 500.40 |

## 7. Conclusions

Deep learning research has significantly advanced cross-modal retrieval, providing elegant solutions and driving substantial progress. In this paper, we presented a comprehensive summary and analysis of numerous notable studies and proposed a taxonomy of cross-modal retrieval mechanisms. We also discussed the challenges and open issues to guide future research from a representation learning perspective. To provide a holistic understanding of the full-cycle methodologies, we covered pre-processing, feature engineering, encoding, cross-modal interaction, decoding, model optimization, and evaluation metrics. Additionally, we employed tables, figures, and equations to enhance the clarity of the primary study.

Despite the extensive efforts, achieving optimal results and precision in cross-modal retrieval remains an ongoing challenge. Key obstacles include feature representation, complex semantic processing, vision language alignment, unified architecture, model optimization, performance evaluation metrics, and the development of more comprehensive datasets.

**Author Contributions:** All authors contributed to the study conception and design. Material preparation, data collection, and analysis were performed by S.W. The first draft of the manuscript was written by S.W.; Data curation, H.M. and S.T.; Writing—review & editing, L.Z. and L.S. All authors have read and agreed to the published version of the manuscript.

**Funding:** This work is supported by the National Key Research and Development Program of China (No. 2022YFC3302103); Fundamental Research Funds for the Central Universities (CUC22GZ052); National Key R&D Program of China (2020YFF0305300).

**Institutional Review Board Statement:** Not applicable.

**Informed Consent Statement:** Not applicable.

**Data Availability Statement:** The data used to support the findings of this study are available from the corresponding author upon request.

**Acknowledgments:** The authors would like to thank all Reviewers and Editors for their comments and views, which improved the quality of this paper.

**Conflicts of Interest:** The authors declare that they have no conflict of interest.

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
