# Peer review of "A Survey of Full-Cycle Cross-Modal Retrieval: From a Representation Learning Perspective"

_applsci, doi:10.3390/app13074571_

Round 1

Reviewer 1 Report

The author does not have to use figures, such as Figure 1 and Figure 2, to show the large amount of existed work related to cross-model retrieval  research. The focus needs to be placed over explanation of  the importance and background of cross-model retrieval related research, instead of giving numbers of publications related to the research topic.

Also the critical part of cross-modal retrieval is the measurement of the content similarity between different types of data. The manuscript provides some paragraphs talking about the universal representation and the unified generation model. But more insightful comparison analysis could be performed to comprehensively discuss the pros and cons of each kind of methods. Tables could be established to show the differences of the methods. 

Reviewer 2 Report

The topic is very timely, needed, and hot. It covers a clear gap, and there is a lot of research going on currently in this area. Therefore, this paper deems an excellent future reference for researchers who work in the area. The overall quality is good and meeting the minimum requirements of the journal. The content is relevant and useful for the researchers' community. However, it has many minor issues that must be fixed properly ranging from literature and lack of suitable directions etc as follows:

1.    The abstract is long and doesn't communicate the problem well. And, it should present the proposed work more clearly.

2.    The abstract must summarise the performance evaluation results and improvement over competitors and/or other solutions.

3. One of the major complaints I have is the poor introduction that fails to motivate the reader to read the paper. This is one of the most important sections of any paper, and I feel that the authors disregard this section compared to an adequate detail in the other sections. Hence, the introduction needs to be rewritten, and expanded in order to motivate the thematic, and the authors must show, while expanding the introduction why it is relevant scientific problem to be solved.

4. Please make sure that all keywords have been used in the abstract and the title.

5- The conclusions section should conclude that you have achieved from the study, contributions of the study to academics and practices, and recommendations of future works.

 6. Thorough proofreading is required

Reviewer 3 Report

Review comments of “A Survey of Full-Cycle Cross-Modal Retrieval: From a Presentation Learning Perspective

Summary

This study presented a survey for full-cycle cross-modal retrieval. The authors investigate the pipeline of cross-modal retrieval from feature engineering, encoding, decoding, pre-training, loss function, benchmark dataset, and evaluation metrics. Specific comments that may help improve this study are listed as follows.

Content Suggestions

1.      What is the definition of “presentation learning” for cross-modal retrieval? What are the unique full-cycle cross-modal retrieval survey results may be achieved from the presentation learning perspective?

2.      Section 5 benchmark datasets. 4. The authors claimed that the quality of dataset plays an important role in the performance of cross-modal retrieval methods and high-quality datasets are able to avoid over-fitting issues during the training of neural networks. However, the performance of models is dependent on the quality of useful features that are extracted from the data, instead of the data itself. The models learn the relationship between the target and these features and thus make predictions. The authors are recommended to clarify what is the definition of the “quality” and “high-quality” to the dataset? Second, the overfitting issue can’t be avoided but be relieved by keeping an optimal tradeoff between model complexity and the amount of dataset. The authors are recommended to clarify the way to avoid over-fitting issue using “high-quality” dataset?

3.      One of the contributions of this survey is to provide a comparative analysis of existing cross-modal retrieval methods including deep learning methods and traditional methods. However, the survey doesn’t provide any performance comparisons between deep learning methods and traditional methods in either image-text or text-image retrieval. Since the commonly used benchmark datasets and evaluation metrics have been respectively presented in Section 5 and Section 6, the authors are recommended to present the comparison results between deep learning methods and traditional methods based on one or more benchmark datasets and evaluation metrics.

4.      Section 2 overview of cross-modal retrieval. The authors are recommended to cite the different deep learning methods (e.g., convolutional neural networks) mentioned in this section and go through the other parts of this survey.

Round 2

Reviewer 3 Report

My comments have been appropriately addressed.